# Detection of Water Spread Area Changes in Eutrophic Lake Using Landsat Data

**DOI:** 10.3390/s22186827

**Published:** 2022-09-09

**Authors:** Vaibhav Deoli, Deepak Kumar, Alban Kuriqi

**Affiliations:** 1Department of Environmental Science and Technology, Indian Institute of Technology, Indian School of Mines, Dhanbad 826004, India; 2Department of Soil and Water Conservation Engineering, GB Pant University of Agriculture and Technology, Pantnagar 263145, India; 3CERIS, Instituto Superior Técnico, Universidade de Lisboa, 1049-001 Lisboa, Portugal; 4Civil Engineering Department, University for Business and Technology, 10000 Pristina, Kosovo

**Keywords:** Landsat, lake, water index, QGIS, remote sensing

## Abstract

Adequate water resource management is essential for fulfilling ecosystem and human needs. Nainital Lake is a popular lake in Uttarakhand State in India, attracting lakhs of tourists annually. Locals also use the lake water for domestic purposes and irrigation. The increasing impact of climate change and over-exploration of water from lakes make their regular monitoring key to implementing effective conservation measures and preventing substantial degradation. In this study, dynamic change in the water spread area of Nainital Lake from 2001 to 2018 has been investigated using the multiband rationing indices, namely normalized difference water index (NDWI), modified normalized difference water index (MNDWI), and water ratio index (WRI). The model has been developed in QGIS 3.4 software. A physical GPS survey of the lake was conducted to check the accuracy of these indices. Furthermore, to determine the trend in water surface area for a studied period, a non-parametric Mann–Kendall test was used. San’s slope estimator test determined the magnitude of the trend and total percentage change. The result of the physical survey shows that NDWI was the best method, with an accuracy of 96.94%. Hence, the lake water spread area trend is determined based on calculated NDWI values. The lake water spread area significantly decreased from March to June and July to October at a 5% significance level. The maximum decrease in water spread area has been determined from March to June (7.7%), which was followed by the period July to October (4.67%) and then November to February (2.79%). The study results show that the lake’s water spread area decreased sharply for the analyzed period. The study might be helpful for the government, policymakers, and water experts to make plans for reclaiming and restoring Nainital Lake. This study is very helpful in states such as Uttarakhand, where physical mapping is not possible every time due to its tough topography and climate conditions.

## 1. Introduction

Lakes are important water bodies assets; therefore, their conservation is crucial in ensuring water security to maintain both ecological and economic benefits [1,2]. A substantial reduction or augmentation of lake areas and their status in water quality can reflect the global environment and climate change impacts [3]. Thus, shrinkage in the water surface areas of lakes is caused by natural phenomena or anthropogenic processes [4,5,6]. Hence, the mappings of surface water bodies are extremely important for policy-making and research purposes.

Remote sensing technology is broadly utilized for surface features mapping, extracting and monitoring the changes, and real-time information acquisition due to its high accuracy and cost-effectiveness [7,8,9,10]. Remote sensing applications in water resources include potential groundwater assessment, change in water bodies areas, flood hazard/damage assessment and management, and water quality assessment and monitoring [11].

So far, various water extraction methods utilizing optical symbolism have been created, which may be graded under four basics types: statistical pattern recognition techniques [12,13], linear unmixing [14], thresholding by a single band [15,16], and spectral indices [17,18]. The spectral index is the most frequently utilized method for simplicity and practicality. Normalized difference water index (NDWI), modified NDWI (MNDWI), automated water extraction index (AWEI), water ratio index (WRI), normalized difference vegetation index (NDWI), enhanced water index (EWI), revised normalized difference water index (RNDWI) and other water extraction indices have been developed and used to extract water features [19,20,21,22,23].

Many recent studies have been conducted on the dynamic changes in water surface area using remote sensing and satellite imagery data [24,25,26,27,28,29,30,31,32,33,34]. Du et al. [35] observed changes in the Qingjiang River basin surface water area from 1973 to 2010 using NDWI and MNDWI. They observed increasing and decreasing trends in the river’s water level through their analysis. Sarp and Ozcelik [36] extracted Burdur Lake of Turkey. They detected the change from 1987 to 2011 using NDWI, MNDWI, AWEI, and support vector machine (SWM). The study concluded that the lake lost almost one-fifth of the water area in 2011 compared to 1987. Wang et al. [37] monitored changes in the water surface area of Ebinur Lake of China using Landsat and Sentinel imageries. They used NDWI and MNDWI to extract the lake. Herndon et al. [38] used Landsat 5–8 series of satellites from 1984 to 2015 to estimate the change in the lake water area. The authors developed fifteen spectral indices to classify the surface water extent. Three simple decision tree methods were developed to identify the surface water in semi-arid environments. Acharya et al. [39] evaluated changes in Phewa, Begnas, and Rupa Lakes of Pokhara, Nepal, using Landsat data within a 25-year gap (1988–2013) using NDWI, MNDWI, and NDVI. The authors suggested that the result might help reclaim and restore the lake. Babaei et al. [40] used Landsat imageries to extract water bodies for Urmia Lake, Iran, by NDWI and MNDWI indices. Cao et al. [41] used cloud-free Landsat imageries to determine the change in the extent of water bodies using Google Earth Engine for the Yellow River of China. The author concluded that the change in river surface area is related to climate change and human activities. Xing et al. [42] traced surface water changes using Landsat images for Shandong Province, China, from 1990 to 2020. The result of the study shows an increasing trend in water bodies. Li et al. [43] extracted surface water using Landsat 8 OLI images using the maximum entropy model.

Uttarakhand is a hilly state of India where the physical mapping of resources is impossible due to its tough topography and climate variability. The periodic study of lake water spread area has not been studied for Nainital Lake. Based on the remote sensing and GIS technique, this is the first study to determine water surface area fluctuation for Nainital Lake. This study attempts to fill this gap by fulfilling the following objectives: (i) to detect the dynamic change in Nainital Lake using different water indices from 2001 to 2018, and (ii) to estimate the accuracy of used water indices by the physical survey of the lake using GPS, and (iii) to estimate the non-parametric trend and magnitude of trend in water spread area of the studied lake.

## 2. Materials and Methods

### 2.1. Study Area

Nainital Lake (Figure 1) is situated in the middle of the Nainital town of Uttarakhand State of India. It is located at an altitude of 1944 m above sea level, at a latitude of 29°23′ N and a longitude of 79°27′. The lake is mango-shaped, approximately 2.5 km in circumference, and surrounded by mountains. The highest are Naina in the north, Deopathaon in the west, and Ayarpathaon in the south. The main water source in Nainital Lake is rainwater and natural base flow. It is fed by twenty water channels, from which only two are perennial open drains [44]. The average annual rainfall of Nainital is 3500 mm, of which 85% is received during the southwest monsoon season [45].

### 2.2. Dataset

Landsat-7 images from 2001 to 2012 and Landsat 8 images from 2013 to 2018 have been used in this study. These images were level-1 from the United States Geological Survey (https://earthexplorer.usgs.gov/) accessed between 2 March 2019 to 10 March 2019 and 15 April 2019. Specifications of these imageries are shown in Table 1.

To estimate the spatiotemporal changes in Nainital lakes, each year is divided into three periods: November to February, March to June, and July to October. The days at which cloud-free Landsat imagery was used in the study every year are in Table 2.

“QGIS 3.4 Madeira” software has created an open-source model for analyzing satellite imagery. The QGIS software can extract water bodies from non-water bodies to calculate land cover area, built-up area, and temperature variations.

### 2.3. Water Body Extraction

The multiband rationing technique has been used to extract the water area from Landsat imageries. Three water indices, namely NDWI, MNDWI, and WRI, have been used to extract the lake water surface area from the image. NDWI, MNDWI, and WRI are based on the spectral index method using QGIS software. NDWI, MNDWI, and WRI were developed primarily to delineate water bodies and enhance their presence in remotely sensed imagery while eliminating soil and vegetation features.

NDWI is primarily used to monitor and detect slight changes in the water content area of the water bodies. Using NIR (near-infrared) and green spectral bands, NDWI can enhance the water bodies in a satellite image. MNDWI uses green and SWIR bands to enhance open water feature extraction. MNDWI can enhance open water features while efficiently suppressing and removing built-up land, vegetation, and soil noise. In the WRI technique, four spectral reflectance bands were used. It is the ratio of the total spectral reflectance of two visible bands (i.e., green and red) to the spectral reflectance of the near-infrared and middle infrared bands (MIR). This method only applies to satellite images with a middle infrared band.

The formulas for the calculation of NDWI, MNDWI, and WRI are described as follows:NDWI = (Green − NIR)/(Green + NIR)(1)
MNDWI = (Green − SWIR_1)/(Green + SWIR_1)(2)
WRI = (Green + Red)/(NIR + SWIR_2)(3)
where Green, NIR, SWIR_1, and SWIR_2 represent the reflection of band 3, band 5, band six, and band 7, respectively, for Landsat 8 images and band 2, band 4, band 5, and band 7 for Landsat 7 images, respectively.

### 2.4. Accuracy Analysis

In remote sensing techniques, physical mapping is important to verify and accurately check satellite data. For this purpose, a survey of the Nainital Lake has been completed using GPS. The GPS survey of Nainital Lake was completed on 20 March 2019, and the area obtained by this survey was compared with the area obtained by the nearest Landsat-8 image on 20 March 2019. The comparison was made to find the error. The best method was determined among NDWI, MNDWI, and WRI by comparing GPS results.

### 2.5. Trend Analysis

To detect the non-parametric trend in hydro-climate data, the Mann–Kendall test [46,47,48,49] has been widely used by researchers and recommended by the World Meteorological Organization [50,51,52,53]. The Sen slope method is used to obtain the magnitude of the trend and percentage change of the lake. Sen’s slope method is a non-parametric method in which slope calculation carries trend prediction [54]. Detailed mathematical expressions have been given in [55,56].

## 3. Results

### 3.1. Comparison of Different Band Ratios for Accurate Water Spread Mapping

The water spread area from 2001 to 2018, from November to February, March to June, and July to October, has been shown in Table 3, Table 4 and Table 5, respectively. A threshold value between −0.05 and 0.9 has been used to estimate the water spread mapping. From Table 3, it could be suggested that the maximum and minimum area obtained by NDWI was 0.464 km^2^ in 2015–2016 and 0.412 km^2^ in 2005–2006, respectively.

This method’s average lake water spread area for the studied period was 0.4406 km^2^ with a standard deviation (SD) of 0.0139 km^2^. The maximum and minimum spread area obtained by the MNDWI method was 0.443 in 2001–2002 and 0.400 km^2^ in 2005–2006, respectively. The average area for the studied period by the MNDWI method was 0.4144 km^2^ with an SD of 0.0102 km^2^. By the WRI method, the maximum and minimum water spread area was 0.449 km^2^ in 2012–2013 and 0.404 km^2^ in 2017–2018, respectively. This method’s average water spread area was 0.4258 km^2^ with SD 0.0125.

Similarly, Table 4 depicted that the average lake water spread area estimated from March to June by NDWI was 0.409 km^2^ with a standard deviation (SD) of 0.01833 km^2^. The maximum and minimum water spread area obtained by MNDWI was 0.423 km^2^ in 2014–2015 and 0.383 km^2^ in 2011–2012, respectively.

The average area for the studied period by the MNDWI method is 0.3998 km^2^ with an SD of 0.0100 km^2^. By the WRI method, the maximum and minimum water spread area is 0.431 km^2^ in the study year 2002–2003 and 0.388 km^2^ in 2017–2018, respectively. This method’s average water spread area was 0.4119 km^2^ with an SD of 0.0124 km^2^.

Further, Table 5 shows Nainital Lake’s calculated water spread area for July to October. The maximum and minimum area obtained by NDWI was 0.467 km^2^ in 2007–2008 and 0.432 km^2^ in 2015–2016, respectively. This method’s average lake water spread area for the studied period was 0.450 km^2^ with a standard deviation (SD) of 0.012 km^2^. The maximum area obtained by the MNDWI method was 0.449 km^2^ in 2008–2009, and the minimum area by this method was 0.405 km^2^ in 2005–2006. The average area for the studied period by the MNDWI method is 0.426 km^2^ with an SD of 0.0132 km^2^. By the WRI method, the maximum and minimum water spread area was 0.479 km^2^ in 2005–2006 and 0.425 km^2^ in 2014–2015, respectively. This method’s average water spread area was 0.4489 km^2^ with an SD of 0.01388.

The water spread area of Nainital Lake was conducted using a GPS survey on 16–17 March 2019, and its results are presented in Table 6. Landsat-8 imagery of 20 March 2019, nearest to the day of the GPS survey, was obtained from earth explorer and then extracted to identify the lake’s area using different multiband water indices.

A comparison between surveyed and estimated water areas by Landsat imagery is presented in Table 6. Table 6 also shows the overall errors in estimating water spread mapping using different water indices. It was also found that the accuracy of all methods used was more than 80%. However, the overall accuracy of NDWI was higher, followed by the WRI method. A 3.06% error or deviation from the original surface water of the lake was noted by NDWI.

In contrast, a 3.16% deviation was noted by using WRI. Due to the higher accuracy, the NDWI method’s areas were considered accurate areas of Nainital Lake in different seasons from 2001 to 2018. The water spread mapping and spatial changes in water spread from 2001–2002 to 2017–2018 of Nainital Lake using NDWI are shown in Figure 2, Figure 3 and Figure 4 for the periods November to February, March to June, and July to October, respectively.

### 3.2. Seasonal Fluctuation in Water Spread Area Using NDWI

Since NDWI estimated the water spread area more accurately in this study than other indices, this rationing technique has been used to analyze water spread fluctuations further. Figure 5 depicts the fluctuations in water spread area from November to February, March to June, and July to October estimated using NDWI. 

Since July to October is a monsoon month in India, the water spread area is expected to be maximum for these months. A remarkable change in water spread could be noticed from March to June. From 2001 to 2018, from July to October, there was a reduction in the water spread area of 0.028154 km^2^ from March to June. Similarly, from November to February, there was a decrease of 0.016846 km^2^. 

### 3.3. Trend and Spatio-Temporal Variability in Water Spread Area

Trend analysis of lake spread area has been conducted using the non-parametric test for 17 years from 2001–2002 to 2017–2018. Due to the unavailability of cloud-free Landsat images, data on the water spread area was unavailable for November to February for the study years 2002–2003 and 2010–2011. Similarly, no data were available for March to June from 2002–2003, 2010–2011, 2012–2013, and 2013–2014. In these cases, linear interpolation has been conducted to determine the trend.

The main results of the trend analysis based on the Mann–Kendall test are presented in Table 7.

The Z-statistics value for November to February was −1.090, March to June was −2.818, and July to October was −2.961. The Mann–Kendall test results indicated a significant negative trend in the water surface area of Nainital Lake for the period March to June and for the period July to October, whereas for November to February, the trend was not significant at a 5% significance level.

The magnitude of the trend and percentage changes of the lake water spread area over the studied period was found using Sen’s slope estimator test. The results have been tabulated in Table 8.

The magnitude of the trend for the period November to February, March to June, and July to October was −0.00070 km^2^/year, −0.00187 km^2^/year, and −0.00123 km^2^/year, respectively. The maximum decrease in water spread area has been obtained for the period March to June (−7.70%), which was followed by the period July to October (−4.67%) and the period November to February (−2.79%).

## 4. Discussion

Accurately estimating surface water bodies becomes essential as water problems increase globally. The important objective of this study is to estimate changes in the surface area of water in Nainital Lake using remote sensing and satellite data. Lake data have been extracted from Landsat images using NDWI, MNDWI, and WRI. Furthermore, a GPS survey of the lake has been completed to estimate the most accurate index for Nainital Lake. Compared to the result of the GPS survey with the area obtained by water indices, the overall accuracy of all used indices is more than 95%. The accuracy of 96.94% when using the NDWI approach compared to 94.31% when using MNDWI and 96.29% using WRI indicates that NDWI is better at extracting small water bodies in the Himalayas region’s water bodies and at separating water bodies from water-like features such as forest shadows. This study’s findings also agree with Rokani et al. [21]. They observed that for modeling temporal lake area change, the NDWI performed better than other indices. Acharya et al. [57] employed the same indices as NDVI, NDWI, MNDWI, and WRI. NDWI can be easily applied to satellite imagery to detect water features from the land surface.

Nevertheless, it should be emphasized that extracting water bodies using a multiband pixel-based method has inherent challenges because they require expertise from analysts to select the accurate training data set [58]. Using a water index such as NDWI is crucial for extracting water bodies to provide information for stockholders, policymakers, and decision-makers. Namely, the developed methodology in this study might be very useful for decision-makers and water managers to take effective measures such as reducing pollution and overwater exploitation from such important and fragile water bodies.

On the other hand, Deoli et al. [59] observed that WRI is the best method to calculate the water extraction area of some Himalayan lakes of India among the studied NDWI, MNDW, WRI, and NDVI indices based on a physical survey of lakes. Furthermore, they calculated the trend in all studied lakes based on Mann–Kendall and Sen’s slope test. The data show a strong downward tendency, and their results align with this study. In addition, Elshabi et al. [60] extract the water surface area using the same water indices. Their overall study accuracy varied from 97.90 to 99.40%, and they recommended these techniques in the areas with the same conditions to extract water bodies. Bijeesh and Narasimhamurthy [61] performed a comparative study to identify water suitability for NDWI, MNDWI, WRI, normalized difference forest index (NDFI), automated water extraction index (AWEI), tasseled cap water index, EWI, and weighted normalized difference water index (WNDWI). All these indices are important for water extraction. All these indices are quick and beneficial in extracting the water bodies. The Mann–Kendall and Sen’s slope estimator shows their efficiency in calculating the trend and magnitude of the trend in hydrological data.

## 5. Conclusions

In this study, the dynamic change in the water spread area of Nainital Lake of India has been completed using the band rationing (i.e., multiband water indices) technique in QGIS software by Landsat images. The results evaluate that the area found by NDWI is very close to the actual area of the lake determined by the GPS survey. The Mann–Kendall test and Sen’s slope were employed to detect a trend in Nainital Lake water spread obtained from NDWI. It has been observed that the water spread area of Nainital Lake shrunk for all studied periods by 0.00070 km^2^/year for the period November to February; −0.00187 km^2^/year for the period March to June, and −0.00123 km^2^/year for the period July to October. The overall decrease in the water spread area of Nainital Lake in the studied years (from 2001 to 2019) was 7.70% for the period March to June, followed by 4.67% for the period July to October and 2.79% for the period November to February. The decreasing lake water area is not a good sign for Nainital Lake. It attracts lakhs of tourists annually and is one of the famous hill stations in India.

Furthermore, the study can also assist policymakers and water information specialists in managing surface water resources. This work will focus on the fusion of the data of studied indices using machine learning techniques to increase efficiency and accuracy. In further work, we will try new indices with machine learning for water surface mapping and monitoring.

## Figures and Tables

**Figure 1 sensors-22-06827-f001:**
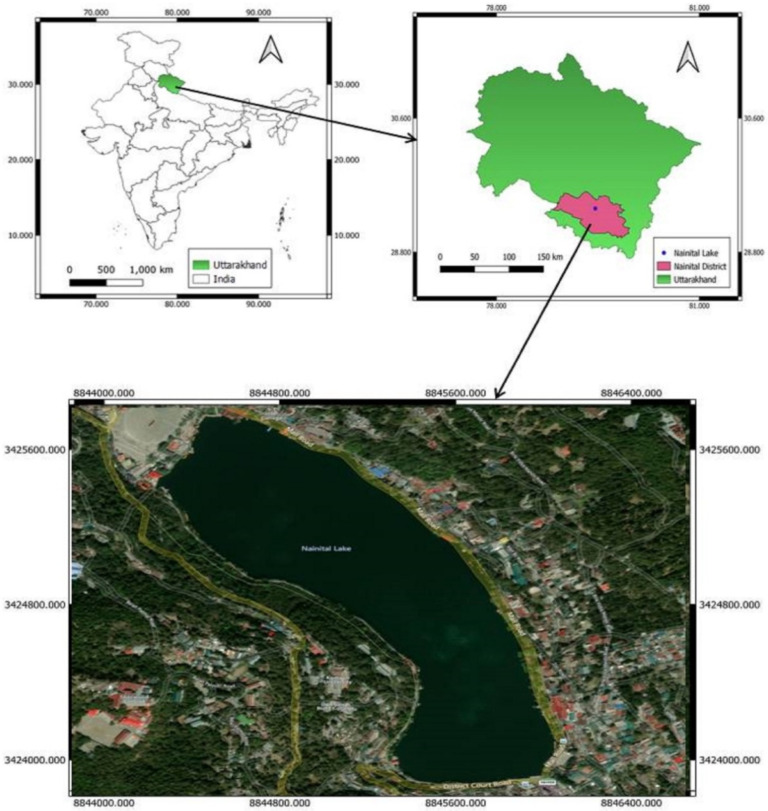
Location Map of Nainital Lake.

**Figure 2 sensors-22-06827-f002:**
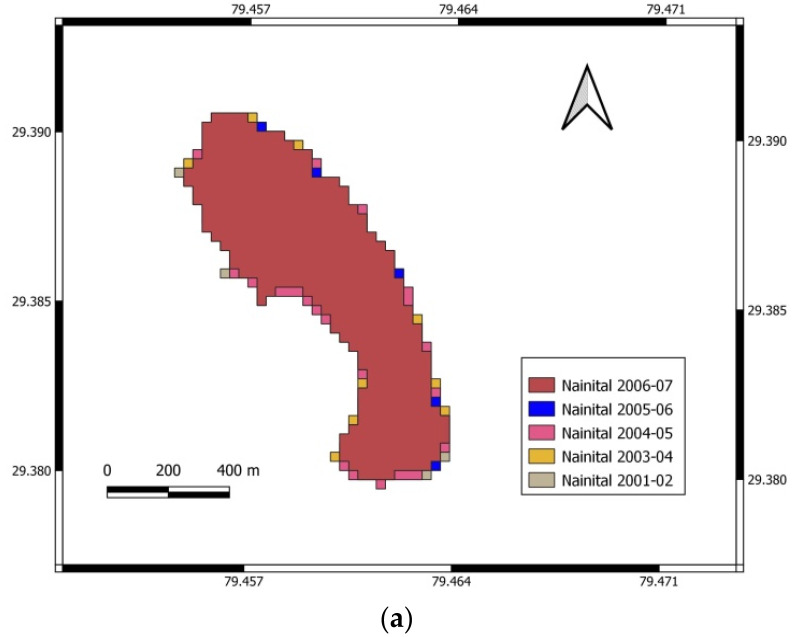
Water spread mapping and spatial changes in water spread from 2001–2002 to 2017–2018 for November to February. (**a**) Variation of water in Nainital Lake from November to February for 2001–2002 to 2006–2007. (**b**) Variation of water in Nainital Lake from November to February for 2007–2008 to 2012–2013. (**c**) Variation of water in Nainital Lake from November to February for 2013–2014 to 2017–2018.

**Figure 3 sensors-22-06827-f003:**
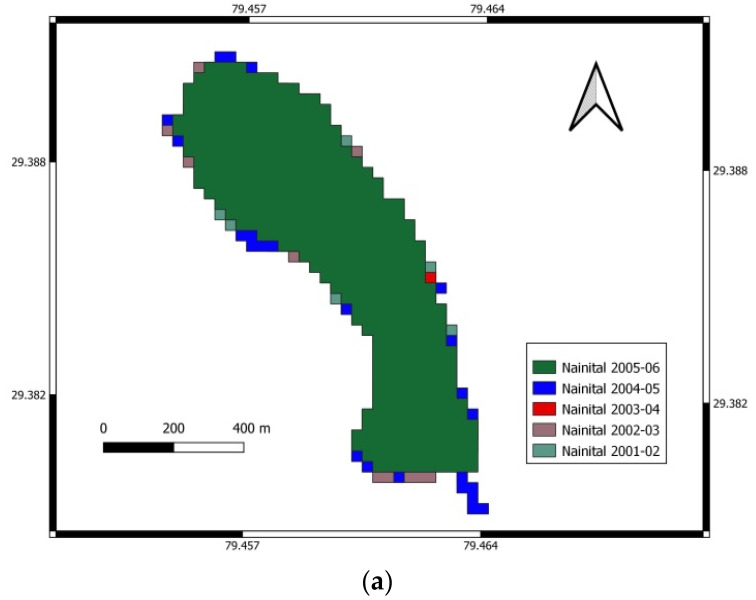
Water spread mapping and spatial changes in water spread from 2001–2002 to 2017–2018 for March to June. (**a**) Variation of water in Nainital Lake from March to June for 2001–2002 to 2005–2006. (**b**) Variation of water in Nainital Lake from March to June for 2006–2007 to 2011–2012. (**c**) Variation of water in Nainital Lake from March to June for 2012–2013 to 2017–2018.

**Figure 4 sensors-22-06827-f004:**
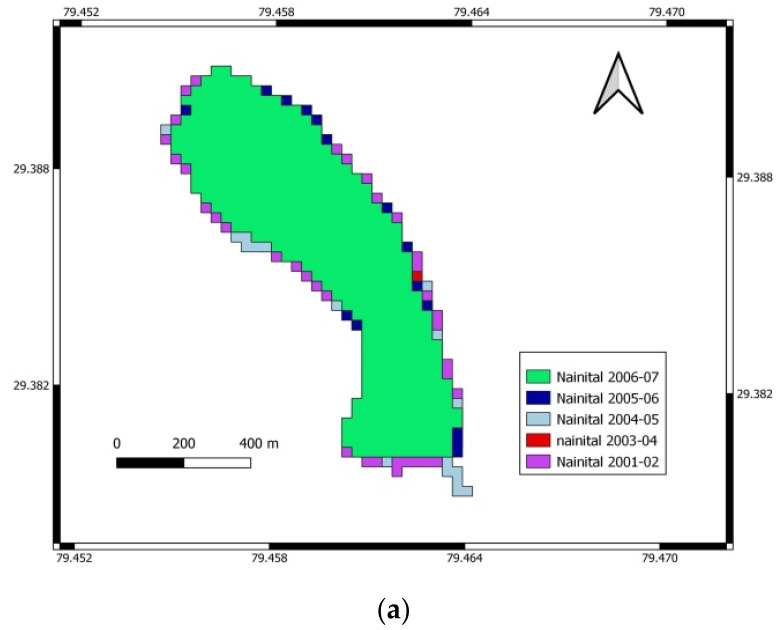
Water spread mapping and spatial changes in water spread from 2001–2002 to 2017–2018 for July to October. (**a**) Variation of water in Nainital Lake from July to October for 2001–2002 to 2005–2006. (**b**) Variation of water in Nainital Lake from July to October for the years 2006–2007 to 2011–2012. (**c**) Variation of water in Nainital Lake from July to October for 2012–2013 to 2017–2018.

**Figure 5 sensors-22-06827-f005:**
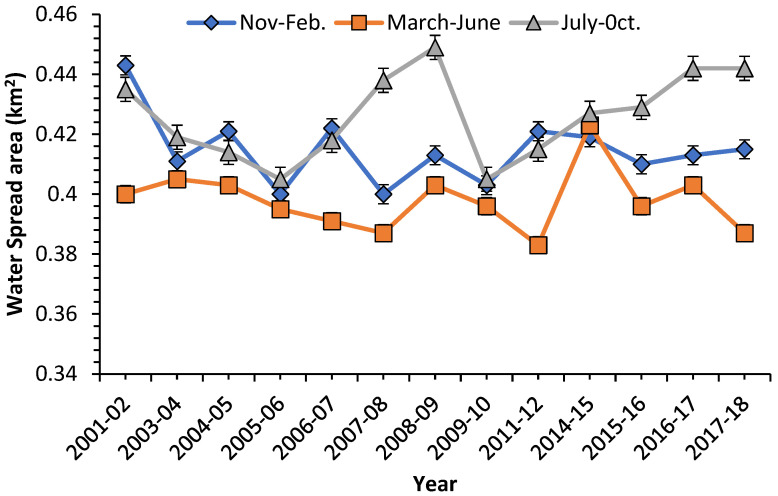
Variation in water spread area of Nainital Lake from 2001 to 2018 (determined by NDWI).

**Table 1 sensors-22-06827-t001:** Specification of Landsat-7 and Landsat 8 bands.

Band Name	Band Number		Resolution (m)
	Landsat 7	Landsat 8	
Deep Blue	-	1	30
Blue	1	2	30
Green	2	3	30
Red	3	4	30
Near Infrared (NIR)	4	5	30
Short-wave Infrared 1 (SWIR 1)	5	6	30
Short-wave Infrared 2 (SWIR 2)	7	7	30
Panchromatic	8	8	15
Cirrus	-	9	30
Thermal Infra-Red 1	6	10	30
Thermal Infra-Red 2	6	11	30

**Table 2 sensors-22-06827-t002:** Cloud-free Landsat imageries were used in this study.

Satellite	Year	Nov–Feb	Mar–Jun	July–Oct
Landsat-7	2000–2001	21-01-2001	27-04-2001	18-09-2001
	2001–2002	23-12-2001	29-03-2002	23-10-2002
	2002–2003	-	01-04-2003	-
	2003–2004	27-11-2003	05-05-2004	28-10-2004
	2004–2005	15-12-2004	09-06-2005	15-10-2005
	2005–2006	04-02-2006	12-06-2006	2-10-2006
	2006–2007	06-01-2007	28-04-2007	21-10-2007
	2007–2008	10-02-2008	30-04-2008	23-10-2008
	2008–2009	27-01-2009	17-04-2009	10-10-2009
	2009–2010	30-01-2010	04-04-2010	29-10-2010
	2010–2011	-	22-03-2011	-
	2011–2012	03-12-2011	27-05-2012	18-10-2012
	2012–2013	22-01-2013	14-05-2013	-
Landsat-8	2013–2014	16-12-2013	10-06-2014	-
	2014–2015	21-02-2015	10-04-2015	03-10-2015
	2015–2016	22-12-2015	12-04-2016	21-10-2016
	2016–2017	10-02-2017	02-06-2017	24-10-2017
	2017–2018	28-01-2018	20-05-2018	27-10-2018

***Note:*** “-” indicates no cloud-free Landsat image available for that duration.

**Table 3 sensors-22-06827-t003:** Water Spread Area of Nainital Lake from November to February.

Year	NDWI	MNDWI	WRI
	Area (km^2^)	Area (km^2^)	Area (km^2^)
2001–2002	0.445	0.443	0.433
2002–2003	-	-	-
2003–2004	0.440	0.411	0.427
2004–2005	0.438	0.421	0.424
2005–2006	0.412	0.400	0.430
2006–2007	0.423	0.422	0.416
2007–2008	0.422	0.400	0.415
2008–2009	0.433	0.413	0.427
2009–2010	0.446	0.403	0.415
2010–2011	-	-	-
2011–2012	0.455	0.421	0.435
2012–2013	0.443	0.415	0.449
2013–2014	0.463	0.411	0.448
2014–2015	0.436	0.419	0.433
2015–2016	0.464	0.410	0.422
2016–2017	0.447	0.413	0.409
2017–2018	0.442	0.415	0.404

***Note*:** “-” indicates no cloud-free Landsat image available for that duration.

**Table 4 sensors-22-06827-t004:** Water Spread Area of Nainital Lake from March to June.

Year	NDWI	MNDWI	WRI
	Area (km^2^)	Area (km^2^)	Area (km^2^)
2001–2002	0.428	0.400	0.421
2002–2003	0.427	0.404	0.431
2003–2004	0.396	0.405	0.416
2004–2005	0.432	0.403	0.428
2005–2006	0.395	0.395	0.422
2006–2007	0.408	0.391	0.420
2007–2008	0.387	0.387	0.401
2008–2009	0.398	0.403	0.403
2009–2010	0.415	0.396	0.406
2010–2011	0.451	0.415	0.445
2011–2012	0.383	0.383	0.406
2012–2013	0.397	0.396	0.410
2013–2014	0.421	0.410	0.408
2014–2015	0.420	0.423	0.402
2015–2016	0.399	0.396	0.403
2016–2017	0.401	0.403	0.393
2017–2018	0.397	0.387	0.388

**Table 5 sensors-22-06827-t005:** Water Spread Area of Nainital Lake July to October.

Year	NDWI	MNDWI	WRI
	Area (km^2^)	Area (km^2^)	Area (km^2^)
2001–2002	0.460	0.435	0.462
2002–2003	-	-	-
2003–2004	0.452	0.419	0.461
2004–2005	0.442	0.414	0.451
2005–2006	0.465	0.405	0.479
2006–2007	0.434	0.418	0.436
2007–2008	0.467	0.438	0.452
2008–2009	0.460	0.449	0.445
2009–2010	0.441	0.405	0.442
2010–2011	-	-	-
2011–2012	0.445	0.415	0.440
2012–2013	-	-	-
2013–2014	-	-	-
2014–2015	0.438	0.427	0.425
2015–2016	0.432	0.429	0.447
2016–2017	0.451	0.442	0.445
2017–2018	0.463	0.442	0.451

***Note*:** “-” indicates no cloud-free Landsat image available for that duration.

**Table 6 sensors-22-06827-t006:** Water spread area using the GPS survey and multiband water indices.

Lake	Using GPS (km^2^)	Using NDWI (km^2^)	Using MNDWI (km^2^)	Using WRI (km^2^)
Nainital	0.457	0.471 (−3.06% deviation from the surveyed area)	0.431 (5.69% deviation from the surveyed area)	0.440 (−3.71% deviation from the surveyed area)

**Table 7 sensors-22-06827-t007:** Trend detection for surface area changes of lakes at a 5% significance level.

Station	Duration	Z-Value	Trend
Nainital	November–February	−1.090	No
	March–June	−2.818	Yes (-) *
	July–October	−2.961	Yes (-) *

* Significant negative trend at 5% significance level.

**Table 8 sensors-22-06827-t008:** The magnitude of change in the water spread area of Nainital Lake is at 5% significance.

Station	Season	Trend Magnitude (km^2^/year)	Change over Study Periods (%)
Nainital	November–February	−0.00070	−2.79
	March–June	−0.00187	−7.70
	July–October	−0.00123	−4.67

## Data Availability

Upon reasonable request.

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
