# Peer review of "Detection of Water Spread Area Changes in Eutrophic Lake Using Landsat Data"

_sensors, 2022, doi:10.3390/s22186827_

Round 1

Reviewer 1 Report

The change of water spread area of Nainital Lake from 2001 to 2018 has been investigated using the multiband rationing indices, NDWI,MNDWI, and WRI, which is compared with a reference GPS method. The result shows that NDWI was the best method, with an accuracy of 96.94%.  This study also implies that the method is very helpful in states like Uttarakhand, where physical mapping is not possible every time due to its tough topography and climate conditions. Although the paper present new findings on potential indices for detecting the water spread area of Nainital Lake, a number of questions are requested to be addressed in the revision before acceptance.

1) The three indices selected should be explained in more details. Why, and What's the new for this study? Are there any previous studies for comparison? For example, NDWI is verified the best.

2) When the GPS method is used for comparison, the authors should answer a very serious question: why not just use GPS method to replace these three indices?

3)  Why is the lake water spread area trend determined with NDWI values, instead of GPS?

Author Response

Reviewer 1

1) The three indices selected should be explained in more details. Why, and what's the new for this study? Are there any previous studies for comparison? For example, NDWI is verified the best.

Answer: Sir, Thanks for reviewing this paper and suggesting improvements. All the indices have been well explained in the paper.

Our study area was the upper Himalayan Region of India, where the regular physical mapping of lakes is always challenging due to its tough topography. This is the first study for the water spread mapping for Nainital Lake by Remote Sensing technique and further trend detection in the lake area. The study is important for Nainital Lake as it attracts lakhs of tourists every year.

Researchers working on remote sensing techniques for natural resources verified their results by ground truthing either by physical mapping or by different accuracy analysis indices like the Kappa coefficient and gave accuracy accordingly.  In this study, we chose a GPS survey of the lake and compared the result with a nearby Landsat 8 image, which is one of the most accurate methods. Based on this, we found NDWI is the best suitable method for this lake.

2) When the GPS method is used for comparison, the authors should answer a very serious question: why not just use GPS method to replace these three indices?

Answer: Thank you for your quarry, sir. In this study, the lake water areas were determined from 2001 to 2018, i.e., for 18 years, to determine whether the lake water area is increasing or decreasing. For this long much of long period, the GPS survey may be a costly affair. Hence we use the Remote Sensing Technique here. As GPS is one of the accurate methods, hence it is used to minimize error by choosing the best method among NDWI, MNDWI, and WRI.

3)  Why is the lake water spread area trend determined with NDWI values, instead of GPS?

Answer: Thank you for your valuable question, sir. The work has been done for using Landsat data for Nainital lake assessment, thus, the trend has also been done using indices used for assessing water spread through remote sensing data. In the present study, NDWI results have been found closer to GPS surveyed data.

Reviewer 2 Report

I enjoyed reading the manuscript.

There are several concerns and recommendations listed below.

-Line150:"After this, the best suitable method among NDWI, MNDWI, and WRI has been obtained for area mapping of Nainital Lake." this is a vague sentence and needs elaboration/justification. 

Why these indices and Landsat8 are selected? explain more in detail by giving examples from the literature of shoreline/coastline detection studies.

For example: https://youtu.be/hJBrbbK5Nyk see two tables in this NASA video (minute 36.18 and 42.07)

This is a shoreline related study as well as water-spread. In the methods part please explain why streoscopy and water line methods and SAR images were not used.

-Fig2 is not enough for assesing the water-spread or coastline change. The reader can be curious about real satellite images at different years to see the change by eyes.

For example Fig 3 in this article

https://doi.org/10.1007/s12665-020-09220-y

Later Figs 5-6 and 8 (in the same article above) will make sense when reading the text.

-Fig3: Will trend analysis with such a small sample (13 points in each line) make sense? Min 30 samples are required for statistical analysis. How do you justify that?

There are different and innovative trend methods. Did you try them?

-L307-310: Why month-to-month changes are reported and not year-to-year changes are not estimated? Is the picture in Fig2 is as scary as Lake/Sea Aral? The reader can not assess this change with only these sub-plots.

The need images from different years.

Author Response

Reviewer 2

  1. Line150:"After this, the best suitable method among NDWI, MNDWI, and WRI has been obtained for area mapping of Nainital Lake." this is a vague sentence and needs elaboration/justification. 

Answer: Thank you for reviewing our paper and valuable comments. The sentence has been changed and written clearly.

  1. Why these indices and Landsat8 are selected? Explain more in detail by giving examples from the literature of shoreline/coastline detection studies. For example: https://youtu.be/hJBrbbK5Nyk sees two tables in this NASA video (minute 36.18 and 42.07). This is a shoreline related study as well as water-spread. In the methods part please explain why stereoscopy and water line methods and SAR images were not used.

Answer: Thank you for your valuable questions. We have added more literature surveys for water body extraction using Landsat Images.

Our manuscript has the title "Detection of Water Spread Area Changes in Eutrophic Lake 2 using Landsat Data". Thus only Landsat data has been used. When we are focusing on the literature review, most researchers have used Landsat data more in place of SAR images for water body extraction. Hence based on the literature survey, we choose the Landsat imagery. Some references are given below:

  • Babaei, H.; Janalipour, M.; Tehrani, N.A. A simple, robust, and automatic approach to extract water body from Landsat images (case study: Lake Urmia, Iran). Journal of Water and Climate Change, 2021, 12(1), 238-249.
  • Cao, H.; Han, L.; Li, L. Changes in extent of open-surface water bodies in China's Yellow River Basin (2000–2020) using Google Earth Engine cloud platform. Anthropocene, 2022, 39, 100346.
  • Xing, W.; Guo, B.; Sheng, Y.; Yang, X.; Ji, M.; Xu, Y. Tracing surface water change from 1990 to 2020 in China's Shandong Province using Landsat series images. Ecological Indicators, 2022, 140, 108993.
  • Li, W.; Zhang, W.; Li, Z.; Wang, Y.; Chen, H.; Gao, H.; Wu, X. A new method for surface water extraction using multi-temporal Landsat 8 images based on maximum entropy model. European Journal of Remote Sensing, 2022, 55(1), 303-312.

In this video (https://youtu.be/hJBrbbK5Nyk), the indices are shown from which NDVI and EVI are used for vegetation cover. In contrast, SAVI, MSAVI, and SATVI are used when work is on soil estimation, and NBR is generally used in the case of a forest fire. Our study is related to water body extraction for an important lake in India. For water body extractions NDWI, MNDWI, and WRI were used.

  1. Fig 2 is not enough for assessing the water-spread or coastline change. The reader can be curious about real satellite images at different years to see the change by eyes. For example Fig 3 in this article  https://doi.org/10.1007/s12665-020-09220-y. Later Figs 5-6 and 8 (in the same article above) will make sense when reading the text.

Answer: Sir, Thank you for your valuable suggestion. We have added all the images for every period. Since we have selected seasonal fluctuations, thus for a particular year, water spread maps for Nov-Feb., March-June and July-Oct. has been added for the study period.

  1. Fig3: Will trend analysis with such a small sample (13 points in each line) make sense? Min 30 samples are required for statistical analysis. How do you justify that? There are different and innovative trend methods. Did you try them?

Answer- The availability of cloud-free images for the study area was one constraint in the present study. Cloud-free LANDSAT images from 2001 to 2018 have been used. Authors have gone through some research articles in which the Mann-Kendall test has been applied for fewer data and the lack of sufficient data.

A few of the research articles are as follows:

  • Neeti, N., & Eastman, J. R. (2011). A contextual mann‐kendall approach for the assessment of trend significance in image time series. Transactions in GIS, 15(5), 599-611.
  • Mohammad, L., Mondal, I., Bandyopadhyay, J., Pham, Q. B., Nguyen, X. C., Dinh, C. D., & Al-Quraishi, A. M. F. (2022). Assessment of spatio-temporal trends of satellite-based aerosol optical depth using Mann–Kendall test and Sen's slope estimator model. Geomatics, Natural Hazards and Risk, 13(1), 1270-1298.

 Authors have also used other techniques, but finally, the Mann-Kendall test has been selected as it also provides the rate at which the trend is increasing or decreasing

  1. L307-310: Why month-to-month changes are reported and not year-to-year changes are not estimated? Is the picture in Fig2 is as scary as Lake/Sea Aral? The reader cannot assess this change with only these sub-plots.

Answer- In the present study, the selection of the study period has been done based on the seasons of a year. In the study area, Nov-Feb is winter, and rainfall is very low compared to other seasons. March-June is the summer season, and July-October is the rainy or monsoon season. So, instead of studying the water spread yearly, it has been done on a seasonal basis. This trend analysis has also been done on a seasonal basis.

The change detection in the water spread area of Nainital lake has been done on pixel to pixel basis. The change detection image of the study lake demonstrates the addition or subtraction of water pixels.

Round 2

Reviewer 1 Report

I'm now satisfied with the answers for the comments and suggest to accept the paper for publication based on minor revision on language and text editing.

Reviewer 2 Report

Thanks for the revised version